# Sex Plays a Multifaceted Role in Asthma Pathogenesis

**DOI:** 10.3390/biom12050650

**Published:** 2022-04-29

**Authors:** Tomomitsu Miyasaka, Kaori Dobashi-Okuyama, Kaori Kawakami, Chiaki Masuda-Suzuki, Motoaki Takayanagi, Isao Ohno

**Affiliations:** 1Division of Pathophysiology, Department of Pharmaceutical Sciences, Faculty of Pharmaceutical Sciences, Tohoku Medical and Pharmaceutical University, 4-4-1 Komatsushima, Aoba-ku, Sendai 981-8558, Japan; akamaco2521@gmail.com (K.D.-O.); fl.nikomako.dr@gmail.com (K.K.); c427.kk.tu.bj@gmail.com (C.M.-S.); takayana@tohoku-mpu.ac.jp (M.T.); 2Center for Medical Education, Faculty of Medicine, Tohoku Medical and Pharmaceutical University, Sendai 983-8536, Japan; iohno@tohoku-mpu.ac.jp

**Keywords:** bronchial asthma, sex, airway epithelial cells, dendritic cells, cluster of differentiation 8^+^ T cells, estradiol, single nucleotide polymorphism, psychological stress, respiratory tract infection, eosinophils

## Abstract

Sex is considered an important risk factor for asthma onset and exacerbation. The prevalence of asthma is higher in boys than in girls during childhood, which shows a reverse trend after puberty—it becomes higher in adult females than in adult males. In addition, asthma severity, characterized by the rate of hospitalization and relapse after discharge from the emergency department, is higher in female patients. Basic research indicates that female sex hormones enhance type 2 adaptive immune responses, and male sex hormones negatively regulate type 2 innate immune responses. However, whether hormone replacement therapy in postmenopausal women increases the risk of current asthma and asthma onset remains controversial in clinical settings. Recently, sex has also been shown to influence the pathophysiology of asthma in its relationship with genetic or other environmental factors, which modulate asthmatic immune responses in the airway mucosa. In this narrative review, we highlight the role of sex in the continuity of the asthmatic immune response from sensing allergens to Th2 cell activation based on our own data. In addition, we elucidate the interactive role of sex with genetic or environmental factors in asthma exacerbation in women.

## 1. Introduction

Sex represents an important biological characteristic in medical science. Increasing evidence has indicated that sex is associated with the prevalence and severity of autoimmune diseases, including rheumatoid arthritis and systemic lupus erythematosus, and allergic diseases, including bronchial asthma [1]. In addition, the male sex is associated with a higher prevalence of non-reproductive cancer and certain types of infectious diseases than that associated with the female sex. Recently, the fatality rate of COVID-19 has been reported to be 1.7-fold higher in males than in females, along with a sex-dependent immune response against coronaviral infection [2].

The global prevalence of asthma has increased since the 1990s, and the global prevalence of conditions diagnosed by doctors, including asthma, clinically treated asthma, and wheezing in adults in 2002–2003 was estimated to be 4.3%, 4.5%, and 8.6%, respectively [3]. The prevalence of asthma in adults is assumed to be 10% and even higher in children in Japan [4], which is somewhat low compared to that in western developed countries [5]. Based on the increasing prevalence of asthma, it is anticipated that 100 million new cases will arise in the next decade [6]. The most recent United States asthma data provided by the Centers for Disease Control and Prevention (CDC) showed that 25 million people suffer from asthma, including 12 million adult (age > 18 years) females and 7.3 million males [7]. The prevalence of asthma after puberty is higher in females than in males in various countries, including Japan, European countries, and the United States [4,7,8]. 

Bronchial asthma is a chronic inflammatory disease of the airways characterized by the accumulation and activation of eosinophils and mast cells in association with increased type 2 helper (Th2) cytokine production [9,10]. Enhanced Th2 cytokine production is attributed to the activation of Th2 cells and type 2 innate lymphocytes (ILC2s). Sensitization to inhalant allergens induces the activation and accumulation of Th2 cells in the airway mucosa via dendritic cell (DC) activation. Additionally, the airway epithelial cell-derived cytokines produced via stimulation with protease activity-comprising allergens activate ILC2s and induce the production of large amounts of interleukin 5 (IL-5) and IL-13 in the absence of antigen-specific stimulation. In addition to type 2 immune responses, an increase in the number of neutrophils in sputum [11] is likely to be associated with increased IL-8 and/or IL-17 levels [12,13] in chronic severe asthma cases. Furthermore, Th1 type immune response, such as interferon-gamma (IFN-γ)-related IP-10 production, may be linked to the severity of asthma in humans via the enhancement of eosinophilic inflammation [14]. 

The pathophysiology of asthma is complex and heterogeneous and is characterized by the interaction of the genetic make-up with the environment [15]. In particular, the influence of sex differences on environmental factors, such as psychological stress receptivity [16], immune responses against viral infection [17], and rhinitis prevalence [18], has been recently elucidated. Moreover, the contribution rate of the specific genetic risk factors for asthma differs between male and female patients [19,20]. This diverse range of asthma pathophysiology is classified as a phenotype or endotype based on clinical characteristics, variety of causation, factors leading to exacerbation, and molecular pathogenesis, including the type of inflammation [21,22]. Clustering analysis showed that the sex-associated phenotype comprises one cluster group, representing late-onset characteristics and decreased lung function associated with obesity [23]. However, the development of the worsening asthma phenotype observed in women is not fully understood.

Sex chromosomes affect the sex-dependent immune responses and disease progression; this is mediated by the production of sex hormones from reproductive organs and the regulation of X chromosome-linked gene expression [24]. The production of sex hormones from the ovaries or testicles affects immune cell fate via genomic and nongenomic pathways by binding with sex hormone receptors [25,26,27]. The X chromosome encodes many immune-related genes such as Toll-like receptor 7 (*TLR7)*, *TLR8*, IL-13 receptor (*IL-13RA)*, cluster of differentiation 40 ligand (*CD40LG*), and forkhead box P3 (*FOXP3*). One of the two X chromosomes in females is silenced; however, it is thought that 15% of genes on the X chromosome avoid this silencing mechanism [24]. The reactivation of X chromosome-linked immune-related genes may be associated with a sex-dependent immune response and disease progression [28]. Asthma has been reported to affect male patients with Klinefelter’s syndrome [29,30], in which the genetic contribution of the X chromosome, including the negative effect of a relatively lower testosterone level, on asthma pathogenesis, is predicted. Thus, although female sex and sex chromosomes may be associated with asthma pathogenesis, the comprehensive mechanisms underlying the worsening asthma phenotype development in women remain unclear.

In the present narrative review, we describe recent developments in research that have examined the critical role of sex and sex-related hormones in the cascade of immune responses from the activation of epithelial cells to eosinophils via innate immune cells and helper T cells in asthma exacerbation. The review emphasizes our data from a murine model of sex-dependent asthma exacerbation. Recent findings have provided evidence of the coordinated contribution of sex to the worsening of asthma. Therefore, we highlight the multifaceted role of sex in the aggravation of asthma and discuss the role of the interplay between sex and genetic factors or other environmental factors in female-predominant asthma exacerbation. 

## 2. Search Strategy and Selection Criteria

In this narrative review, we have discussed the relationship between sex and genetic factors or environmental factors based on the literature written in English that could be picked up from search results in PubMed and CDC reports. This review covered articles written between 2000 and 2022 and some important literature written in the 1960s and 1990s. Relevant literature was searched in PubMed using various combinations of the following keywords: “asthma”, “exacerbation”, “onset”, and “sex” or “gender”; plus either of the following: “prevalence”, “severity”, “pathophysiology”, “phenotype”, “psychological stress”, “comorbidity”, “ozone”, “air pollution”, “vitamin D”, “infection”, ”rhinitis”, “genetic factor”, “single nucleotide polymorphism”, “chromosome”, “estradiol”, “progesterone”, “testosterone”, “hormone”, “immune response”, “Th1”, “Th2”, “Th17”, “innate lymphocyte”, “airway epithelial cell”, “macrophage”, “monocyte”, “dendritic cell”, and “eosinophil”. Therefore, the description might be biased depending on how we selected the keywords and combined them. This review does not include the grey literature (Google Scholar) and ongoing studies in clinical trial registries. Furthermore, we excluded case reports as much as possible, wherein observational studies, randomized studies, and controlled trials were included. In the case of conflicting opinions reported in the field, we assigned priority to more recent publications, meta-analyses, and systematic reviews. Furthermore, we gave priority to the results from human studies over animal studies. In human studies, we described the ages of participants in the text as appropriate to interpret the results. 

## 3. Sex Influences the Clinical Outcome and Phenotype of Asthma

The prevalence and frequency of asthma symptoms are higher in boys than in girls; however, these trends are reversed after puberty and sexual maturation [31,32,33]. The annual trend of hospital admission from 2001 to 2010 in the United States showed that 66.6% of admitted patients were female in total asthma admissions, and females accounted for the majority of admissions over the years [34]. A population-based cohort study on 209,054 individuals with asthma (age ≥ 66 years) using health administrative data in Canada showed that female patients have lower rates of spirometry but higher rates of asthma-specific emergency department (ED) and physician office visits and asthma reliever medication than male patients [35,36]. In addition, female patients promptly relapse after discharge from the ED [37], and the hospitalization rate is higher in adult females than in adult males with asthma exacerbations that visit the ED [38]. Female patients visiting the ED experience more severe symptoms than male patients [39]. In addition, hospitalization costs increase with age, with females reporting higher expenses than males [40]. 

Clustering analyses have revealed the characteristics of the clinical phenotypes of sex-related asthma. In adult-onset asthma cases, sex-related phenotypes are defined as the predominance of female patients with obesity and high symptom expression in the absence of eosinophilic airway inflammation [41]. Moore et al., identified the characteristics of the female-predominant asthma phenotype as a distinguishing cluster mainly comprising women (mean age, 50 years; range, 34–68 years) who are less likely to be atopic with the highest body mass index at late-onset and have decreased baseline pulmonary function [42]. A Taiwanese adult asthma cohort study on patients with mild-to-severe asthma (age > 20 years) identified two female-specific asthma phenotypes: atopy with eosinophil-predominant phenotype and obesity with neutrophil-predominant phenotype [43]. Another recent clustering analysis of puberty-onset asthma reported a female-dominant atopic asthma group with a low lung function phenotype [44]. Thus, while the phenotype of obesity with a high symptom expression and low lung function in female patients has become evident, patient groups may show low lung function with atopic and enhanced eosinophil inflammation in relatively young women after puberty.

## 4. Sex Hormones Influence the Pathophysiology of Asthma

Alterations in female hormone levels along with sexual maturation, menstrual cycle, pregnancy, or hormone replacement therapy (HRT) are associated with sex-related differences in an increased risk for asthma [45], asthma-related hospital admission [46], and the prevalence of asthma-like symptoms [47,48]. Ban Al-Sahab et al., showed that the onset of early menarche is associated with a >2-fold higher risk of developing asthma during early adulthood than that in girls who mature at an average age [49]. Furthermore, compared with pre-menopause, surgical menopause—but not natural menopause—is associated with an increased risk of asthma onset [50]. In an analysis of the Nurses’ Health Study on naturally menopausal women, the age-adjusted relative risk of asthma in individuals with current and past use of postmenopausal hormones is 1.49–1.52-fold higher than that in those without hormone use. The study has indicated that estrogen plays an important role in the pathogenesis of asthma and that long-term and/or high-dose use of postmenopausal hormone therapy increases the subsequent risk of asthma [51]. Prospective analysis of cohort studies also confirmed that the risk of asthma onset is higher in individuals using postmenopausal hormones compared with that in individuals who never used hormones [52,53]. In particular, the increase in the risk of asthma onset is significant only among women who use estrogen alone. Systematic reviews and meta-analyses have reported that the association between menopause and asthma rates is consolidated in postmenopausal HRT users compared with that in premenopausal women reporting HRT use, whereas the association between menopause and asthma rates is not significant in non-users of HRT [54,55]. Regarding asthma exacerbation, the use of HRT, including combined therapy and estrogen alone, is associated with an increased risk of severe asthma exacerbation in perimenopausal/postmenopausal women with established asthma [56]. 

Apart from the positive correlation, conflicting evidence has been reported regarding the association of menopause and HRT with the risk of asthma. Few studies showed that HRT in postmenopausal women increases pulmonary function, including forced expiratory volume in one second (FEV1), forced vital capacity (FVC), forced expiratory flow 25–75%, and peak expiratory flow rate, and also induces less obstruction [57,58,59]. Furthermore, HRT users are less likely to show bronchial hyper-responsiveness than non-users [60], and HRT reduces the risk of late-onset asthma development in menopausal women [61]. A northern European population study indicated that the risk of new-onset asthma increases in postmenopausal women during the transitional period of the postmenopausal state from early postmenopausal to the late postmenopausal state compared with that in non-menopausal women [62]. This study also suggested that the differences in study populations, definition of exposure, and study outcomes yield conflicting results regarding the effect of menopause on asthma. Similarly, Shah et al., reported that conflicting evidence in previous research regarding the effect of HRT on asthma pathogenesis could be attributed to the differences in the study population, collection method of data, duration of HRT, and update period of information [61].

## 5. Influence of Sex on the Continuity of Immune Responses from the Sensing of Aeroallergens by Airway Epithelial Cells to Eosinophil Activation in Asthma

Typical asthmatic features, such as enhanced airway hyperresponsiveness and airway inflammation, are mainly mediated by Th2-type cytokines [63]. In addition, Th9- and Th17-type immune responses drive mucus production and induce airway smooth muscle contraction with the induction of neutrophilic inflammation, respectively [64,65]. Furthermore, Th17-type inflammation in asthma is associated with the development of steroid resistance [13]. A clinical study demonstrated that a greater number of IL-13-producing T cells [66] and IL-17A-producing CD4^+^ memory T cells [67] and an increased proportion of IL-5-producing ILC2s [68] are present in the peripheral blood of female patients with asthma compared to those in the peripheral blood of male patients. These clinical data suggest that symptom-based features of worsening asthma in female patients result from sex-related differences in immune responses associated with asthma.

### 5.1. Sex-Related Differences in the Activation of Airway Epithelial Cells: Initiation of Allergic Immune Responses

The airway epithelium represents the first defense barrier that prevents the invasion of aeroallergens into the respiratory tract mucosa, in which tight junction factors, including claudins, occludin, and zonula occludens, create continuous cell–cell contacts [69]. In addition, the ciliary beating of ciliated epithelial cells mediates the clearance of inhaled allergens and foreign matter. In asthma, airway inflammation caused by tumor necrosis factor (TNF)-α, IFN-γ, IL-4, and IL-13 dysregulate epithelial barrier activities [70]. Although the exact role of sex in this physical defense mechanism remains unclear, it is possible that the female-predominant IL-13 production observed in patients with asthma may be associated with the defective functional activity of tight junction proteins [71]. In addition, data from an asthma mouse model indicated that airway epithelial barrier dysfunction caused by prenatal stress during the development period of the lungs may be responsible for asthmatic features perpetuated in adolescent females [72]. In addition, Jain et al., reported that airway epithelial cell exposure to progesterone (PR) decreases the ciliary beat frequency, suggesting that sex hormones influence the function of a key component of the mucociliary apparatus in epithelial cells [73]. Since the receptors for estrogen α (Erα) and β (Erβ), PR, and androgen are expressed in human bronchial epithelial cells [73,74,75], further studies regarding the role of sex hormones in the physical barrier function of airway epithelial cells are required. 

In addition to the physical defense mediated by epithelial cells, the airway epithelial cells function as initial sensors for detecting allergens via pattern recognition receptors. After sensing allergens, the airway epithelial cells produce cytokines, such as IL-25, IL-33, and thymic stromal lymphopoietin (TSLP), and chemokines, such as C-C motif chemokine (CCL)2 and CCL20 [76]. These immune responses in airway epithelial cells initiate subsequent innate and adaptive immune responses in asthma via the induction and activation of immune cells in the airway mucosa [77]. Our investigation demonstrated sex-related differences in cytokine and chemokine production from Ep-CAM^+^ airway epithelial cells in asthmatic mice. In an ovalbumin (OVA)-induced asthma mouse model, two rounds of sensitization using OVA/aluminum hydroxide and subsequent OVA inhalation induced greater IL-33 and CCL2 production by epithelial cells from female mice at 1 and 4 h after OVA inhalation, respectively, than that in male mice (Figure 1A, a detailed methodology for Figure 1 is described in Appendix A). In the sex-dependent difference in cytokine and chemokine production from human airway epithelial cells, we also demonstrated enhanced production of CCL2 and granulocyte-macrophage colony-stimulating factor from BEAS-2B cells, which was dose-dependently induced by 17β-estradiol treatment in the presence of TNF-α (Figure 1B). Our data further showed that the supernatant from BEAS-2B cells stimulated with 17β-estradiol plus TNF-α increased the proportion of CD86-expressing CD1c^+^ cells among human CD14^+^ cells in vitro compared to that obtained using BEAS-2B cells stimulated with TNF-α alone. This suggested that the enhanced maturation factor production from epithelial cells induced by 17β-estradiol treatment may be responsible for the increased number of activated DCs in women (Figure 1C). In concordance with this observation, bronchoalveolar lavage (BAL) fluid from female mice after OVA inhalation induced a higher *Cd86* mRNA expression in splenic CD11c^+^ cells than those from male mice (Figure 1D). Previously, it was reported that estradiol-dependent signals using Erα induce enhanced mucus production [78], nitric oxide production [79], and IL-33 production from airway epithelial cells [80]. Zein et al., reported that androgen receptor expression levels in epithelial cells are positively associated with the FEV1/FVC ratio and the total Asthma Quality of Life Questionnaire score, but negatively associated with fractional exhaled nitric oxide and nitric oxide synthase gene expression [81]. These previous studies suggest that female sex hormones enhance epithelial cell differentiation into goblet cells and immune responses that aggravate asthma, while the effect of male sex hormones on airway epithelial cells improves asthma symptoms.

### 5.2. Sex-Related Differences in the Activation of DCs

DCs represent the most proficient antigen-presenting cells and bridge the gap between innate and adaptive immune responses by priming helper and cytotoxic T cells in response to respiratory allergens, which is a critical step of atopic responses in the development and exacerbation of asthma [82]. In humans, classical DCs, blood DC antigen 1 (BDCA1^+^) DC and BDCA3^+^ DC subsets, are involved in the pathogenesis of asthma. While the allergen challenge increases the proportion of BDCA1^+^ myeloid DCs (mDCs) in BAL fluids [83] and BDCA3^+^ mDCs in sputum [84] of patients with asthma, the proportion of DCs in the blood reduces significantly at 3–6 h after allergen inhalation [85]. Animal studies have verified the role of CD11b^+^ CD103^−^ (CD11b^+^) and CD11b^−^ CD103^+^ (CD103^+^) DCs, which are homologs of human CD1c (BDCA1)^+^ and CD141 (BDCA3)^+^ DCs, respectively, in asthmatic immune responses [86,87]. However, the main DC subset responsible for asthmatic immune response remains unclear. Hoffmann et al., suggested that the nature of allergens such as OVA, cockroaches, or house dust mites (HDMs) and the amount involved during the sensitization phase may explain the difference in the DC subset associated with asthma [88].

Our data using the OVA-induced asthma mouse model showed a larger proportion of CD11b^+^ DCs and CD103^+^ DCs in the bronchial lymph nodes (BLNs) of female mice than that in male mice. In addition, antigen uptake ability, migration capacity, costimulatory molecule expression levels, and Th2 cell-inducing capacity in CD103^+^ DCs in BLNs are significantly greater in female mice than those in male mice [89]. Notably, 17β-estradiol treatment increased CD86 expression on DCs in our experiment. Moreover, when we blocked the interaction of CD86 on DCs using the specific antibodies in IL-5 production from CD4^+^ T cells in co-culture experiments, the sex-related difference in IL-5 production between male and female mice was completely abolished. These results indicated that the enhanced CD86 expression on DCs caused by 17β-estradiol is responsible for the enhanced Th2 cell activation in female mice. Interestingly, the 17β-estradiol-dependent enhancement of CD86 and major histocompatibility complex-II expression is further enhanced upon IL-33 treatment [89]. Together, it can be inferred that treatment with female sex hormones increases not only the ability of CD103^+^ DCs to differentiate Th2 cells by themselves but also emphatically enhances DC function in cooperation with IL-33 surrounding DCs. Xiu et al., demonstrated the estradiol-dependent DC differentiation and activation in vitro; however, PR treatment reduces them dose-dependently. In addition, the cotreatment of PR and estradiol significantly suppresses DC activation [90]. These data suggest the distinctive roles of female sex hormones in the pathophysiology of female asthma, and their balance in the menstrual cycle may control the nature of DCs in asthma.

### 5.3. Sex-Related Differences in Helper T Cell Responses in Asthma

The pathogenesis of asthma is mainly mediated by type-2 inflammation, with increased production of Th2 cytokines such as IL-4, IL-5, and IL-13 by Th2 cells and ILC2s [91]. In addition, IL-17-mediated inflammation with increased neutrophil infiltration appears to be associated with steroid resistance phenotypes in asthma [13,65]. A few studies have demonstrated sex-related differences in the cytokine production capacity of T cells in patients with asthma. Loza et al., reported that the rate of increase in the proportion of IL-13^+^ T cells in peripheral blood lymphocytes after stimulation with CD3/CD28 plus IL-2 ex vivo is significantly greater in female atopic adult patients with asthma than in male patients [66]. Furthermore, the proportion of IL-17A^+^ memory Th17 cells is increased in female patients with severe asthma than in male patients, while the proportion of IL-4^+^ memory T cells is similar in female and male patients with severe asthma [67].

Erα, Erβ, PRα, and PRβ are expressed in human T cells [92,93]. Stimulation with estradiol and/or PR increases IL-4 production and GATA-3 expression in human peripheral blood mononuclear cells [94]. A similar trend was observed for IL-17A production in asthmatic mice. Hormone replacement with estradiol and PR significantly increased IL-17A production from Th17 cells in ovariectomized mice [67]. In particular, signaling via Erα is involved in IL-23 receptor expression and has been shown to be responsible for IL-17 production and Th17 cell differentiation [67]. Furthermore, in a mouse model of HDM-induced asthma, gonadectomy in female mice significantly reduced the proportion of IL-13^+^ and IL-17A^+^ T cells; in contrast, gonadectomy in male mice significantly increased the proportion of IL-13^+^ and IL-17A^+^ T cells [95].

The production of larger amounts of Th2 cytokines by helper T cells in female mice than that produced by male mice is not understood. Clinical studies on HRT in female patients suggested that female sex hormone-induced worsening or alleviation of pulmonary function remains controversial. Therefore, we evaluated the role of CD8^+^ T cells in female-predominant cytokine production from CD4^+^ T cells, focusing on the dual anti- and pro-inflammatory roles of CD8^+^ T cells in the asthmatic immune response [96]. In particular, the cytokine-producing capacity of CD8^+^ T cells is similar to that of CD4^+^ T cells, which impacts the modulation of Th1 or Th2 polarization of CD4^+^ T cells [97]. Our data demonstrated that the increased number of CD4^+^ T cells in BAL fluid and Th2 cytokine production in the lungs of female mice were reversed in CD8-disrupted mice, suggesting that CD8^+^ cells are involved in sex-related asthmatic immune responses [98]. CD8^+^ T cells derived from male mice secreted greater amounts of IFN-γ than those derived from female mice. Furthermore, inoculation of CD8^+^ T cells derived from male mice, but not those from female mice, into sensitized mice suppressed eosinophil accumulation in BAL fluid in vivo. Similarly, co-culture of CD8^+^ T cells derived from male mice, but not from female mice, with CD4^+^ T cells in the presence of CD11c^+^ cells suppressed IL-4 production from CD4^+^ T cells in vitro. Notably, CD4^+^ T cells derived from male mice expressed greater amounts of IFN-γ receptors α and β than the female mouse-derived CD4^+^ T cells. Furthermore, CD4^+^ T cells in the BLNs of male and female mice produced comparable levels of IL-4 ex vivo. Nevertheless, in the presence of recombinant IFN-γ, CD4^+^ T cells derived from female mice produced greater amounts of IL-4 compared with that produced by CD4^+^ T cells derived from male mice in vitro. Thus, the surrounding environment of CD4^+^ T cells, such as the coexistence of CD8^+^ T cells and sex-related differences in the sensitivity to IFN-γ of CD4^+^ T cells, orchestrate the female-predominant IL-4 production from CD4^+^ T cells.

### 5.4. Sex-Related Alteration of Macrophages, Monocytes, and ILC2s

Several reports have indicated that lung macrophages contribute to asthma development [99,100,101]. Among macrophage subsets, M2 macrophages aggravate asthma by enhancing Th2 inflammation [102]. Moreover, the number of M1 macrophages has been linked to asthma severity and poor response to corticosteroids [103], even though M1 macrophages are associated with Th1 inflammation [104]. A greater number of activated macrophages has been found to influence mDC migration and airway inflammation in the lungs of asthmatic female mice than in asthmatic male mice [102]. In the OVA and IL-33-treated asthma mouse model, enhanced activated macrophage differentiation is induced in females compared with that in male mice, along with a mechanism dependent on IL-13 production and the signal transducer and activator of transcription 6 activation [105]. In addition, sex-related differences in the development of the alveolar macrophage phenotype, but not ILC2 signaling, are responsible for the female bias in eosinophilic inflammation in the airways after multiwalled carbon nanotube exposure [106]. Draijer et al., demonstrated that patients with asthma have a higher proportion of IRF5^+^ M1 type macrophages and CD206^+^ M2 type macrophages in bronchial biopsies than that in healthy individuals. In particular, male patients with asthma have a relatively greater number of IRF5^+^ M1 macrophages, and female patients with asthma have a greater number of CD206^+^ M2 macrophages [107]. 

Zasłona et al., indicated that circulating monocyte populations are putative precursors of macrophages that promote asthmatic inflammation in the lung, whereas resident alveolar macrophages attenuate allergic inflammation [108]. Regarding monocytes derived from human peripheral blood, Becerra-Díaz et al., recently demonstrated that the expression of the common γ chain, a part of the IL-4 and IL-13 receptors, is enhanced in female patients with asthma, and monocytes from these females express a larger amount of CX3C chemokine receptor 1 than that in male patients with asthma. This study suggests that macrophage survival is enhanced in female patients [109]. Monocyte-derived macrophages from women express higher levels of matrix metalloproteinase 12 and transglutaminase 2, suggesting that women show female-predominant potency of M2-type macrophage differentiation [109]. 

ILC2s are activated by several factors, such as Th2 cytokines, epithelial cell-derived cytokines, lipid mediators, and cell–cell signaling [110], and are responsible for the innate phase of immune reactions in asthma [111]. In a clinical study, Cephus et al., demonstrated that the proportion of IL-5^+^ ILC2s is increased in patients with moderate-to-severe asthma compared to that in healthy controls, and female patients have an increased proportion of circulating IL5^+^ ILC2s compared to that in male patients [68]. Data from an experimental asthma mouse model treated with *Alternaria alternata* extract showed that testosterone is responsible for decreasing the expression of IL-33 and TSLP, followed by the reduction of Alternaria extract-induced IL-5^+^ and IL-13^+^ ILC2 populations and the proportion of lung eosinophils [68]. ILC2 progenitor cells primarily express androgen receptor genes, whereas the expression of ER genes (*Esr1* or *Esr2*) is almost undetectable. In fact, androgen receptor signaling plays an important role in sex dysmorphism of the asthmatic innate immune response by negatively regulating the homeostasis, proliferation, and function of ILC2s [112,113].

### 5.5. Sex-Related Alteration of Eosinophils

Enhanced amounts of GM-CSF and IL-5, an eosinophil accumulation and activation factor, followed by an increased number of eosinophils in airways, were observed in asthmatic female mice. In addition, β-estradiol directly enhances the adhesion and degradation of eosinophils, while testosterone suppresses their adhesion and viability [114]. Moreover, the pharmacological antagonistic effect of ER reduced the number of eosinophils in the lung of intact sensitized mice [115]. These results obtained from in vivo and in vitro studies suggest that eosinophil function in asthma may be directly and indirectly affected by sex. However, the sex-related differences in eosinophil counts in humans via a stratified analysis of asthma pathophysiology have not yet been deciphered.

## 6. Cross-Interaction of Sex with Genetic Factors or Environmental Factors in Asthma Exacerbation

The asthma phenotype is generated by the interaction of genetic and environmental factors [116]. Although the role of gene-environment interactions in the sex-related asthma phenotype is not clearly understood, it has become increasingly clear that genetic factors such as single nucleotide polymorphisms (SNPs) of asthma-related genes may affect sex-related modulation of cytokine production. Furthermore, sex has been shown to be associated with other environmental risk factors such as psychological stress, viral infection, and upper respiratory tract inflammation, which exacerbate asthmatic immune responses and signs.

### 6.1. Interaction of Sex with the Genetic Make-Up

The pathophysiology of asthma is characterized by a complex relationship between the genetic background of individuals and the environment of the patients [15]. This concept suggests that asthma pathogenesis results from the interaction of a number of factors, and not a single factor. Genome-wide interaction studies identified six sex and ethnicity-specific asthma risk alleles in the regions of *IRF1*, *RAB11FIP2*, *AK057517*, *ERBB4*, *C6orf118*, and *RAP1GAP2* [117]. Furthermore, another genome-wide analysis employing 411 European American asthmatic cases and 297 non-asthmatic cases suggested several sex-related candidate SNPs in gene loci including *TSLP* [118]. In a study conducted in a Pakistani population, the TT genotype in rs2583476 SNP of *FCER1B* gene was more prevalent in male patients with asthma than in healthy male controls, while the CT genotype in rs11650680 SNP of *ORMDL3* gene was more prevalent in female asthma patients than in healthy female controls [119]. Further detailed biological effect on the association between sex and SNPs in *TSLP* has been reported. While the T allele of the *TSLP* SNP rs2289276 is significantly associated with a reduced risk of asthma in females, the T allele of rs1837253 is associated with a reduced risk of asthma in males [19]. Ranjbar et al., similarly reported the association of sex with *TSLP* SNPs; the T/T allele of rs2289276 is inversely associated with the risk of asthma in females, but not in males, in an Iranian population [20]. Compared with the C/C SNP of the *TSLP* rs1837253, the T/T SNP has been reported to significantly reduce TSLP production from nasal epithelial cells upon stimulation with PolyI:C in vitro [120,121]. These data indicate that sex represents a risk factor in asthma pathogenesis and is associated with differences in the genetic backgrounds of patients.

### 6.2. Interaction of Sex with Psychological Stress

Psychological stress is recognized as a risk factor for asthma exacerbation [122]. Approximately 47.9% of the Japanese population aged >12 years, except for hospitalized patients, suffer from psychological stress. In the psychological stress signal transduction pathway from the brain to the lungs, psychological stress initially induces brain activity, followed by the efflux of neuropeptides such as opioids, which activate the hypothalamic-pituitary-adrenal axis, the autonomic nervous system, and the parasympathetic nervous system [122]. As a result of the activation of these axes, stress hormones, such as glucocorticoids, epinephrine, norepinephrine, and acetylcholine, are released and then amplify Th2-type inflammation in the lungs. Depression is twice as common in women than in men, in which sex differences in the living environment in daily life and female hormone levels may be involved and amplify the neuropsychiatric responses [123]. A questionnaire-based survey of 3,085 patients with asthma demonstrated that psychological stress is involved in 10.5–15.3% of asthma exacerbations, and the contribution rate is significantly higher in female patients than in male patients [124]. Regarding the sex-related cortisol response to psychological stress, Lovallo et al., demonstrated the possibility that women have cortisol stress responses that are more heavily regulated by endogenous opioid mechanisms than that in men, and alteration of the affinity of the μ-opioid receptor caused by SNPs may affect women to a different extent than men [16]. Our previous study using an asthma mouse model showed that psychological stress such as 6 h restraint per day for 3 consecutive days and 3 min forced swim per day for 4 consecutive days increases eosinophil counts in BAL fluids in female mice than those in male mice [125]. This evidence confirmed that females are more sensitive to psychological stress than males, which in part contributes to female-predominant asthma exacerbation. 

### 6.3. Interaction of Sex with Respiratory Infection or Upper Airway Inflammation

The modulation of cytokine and chemokine production by epithelial cells plays an important role in the initiation of allergic immune responses, as described in detail in the previous section. Previous studies have shown that respiratory syncytial (RS) virus infection induces an increase in TSLP production [126], which activates IL-13-producing ILC2s [127]. In asthma pathogenesis before puberty, asthma prevalence is higher in boys than in girls before puberty, and the role of RS virus infection in immunomodulation has garnered attention. Malinczak et al., demonstrated that better viral control along with correlated expression of interferon-β is observed in female mice, but not in male mice [17]. This sex dysmorphism in response to RS virus infection is associated with enhanced *Tslp* and *Il33* mRNA expression, which allows infiltration of innate immune cells into the lungs and Th2 and Th17-skewing in male mice compared with that in female mice. Thus, the sex-dependent modulation of epithelial cell-derived cytokines by RS virus infection may induce sex-dependent asthma in early life.

The “one airway, one disease” concept takes hold in light of Th2 inflammation [128]. Chronic rhinosinusitis and rhinitis induce mouth breathing, which allows dry air to flow into the lower respiratory tract and induces contraction of airway smooth muscle and bronchi. In addition, the release of mediators and cytokines in the upper airway in response to the stimulation of foreign antigens further amplifies the inflammation of the lower respiratory tract. Thus, when considering sex-related differences in lower respiratory inflammation in asthma pathogenesis, the effects of sex-related differences in upper airway inflammation must be accounted for. Previously, a review reported the female predominance in the prevalence of rhinitis in patients aged 11–18 years [18]. Especially, data from six European population-based birth cohort studies indicate that the male predominance in allergic prevalence before puberty and the “sex-shift” towards females after puberty onset are strongest in multimorbid patients with asthma and rhinitis [129]. Thus, although the effects of upper airway inflammation on sex-related differences in asthma prevalence have been reported, further evidence is required to directly link sex differences in upper and lower respiratory immune responses.

### 6.4. Interaction of Sex with Ozone or Air Pollution in Asthma

Ozone and air pollution are known as asthma triggers, and several studies have shown sex-related differences in ozone or air pollution-induced asthma pathogenesis. A study involving 30,139 Chinese children aged 3–12 years demonstrated that the effects of air pollutants such as PM10 and SO_2_ on asthma were stronger in male children than those in female children without an allergic predisposition [130]. In contrast, among children with an allergic predisposition, more positive associations between air pollutants such as SO_2_, NO_2_, and O_3_ and the prevalence of clinically diagnosed asthma were higher in female children than that in male children [130]. Similarly, Glad et al., demonstrated that the relationship between ozone/PM2.5 levels and emergency department visits for asthma are influenced by gender [131]. These sex-related differences in the airway response to ozone and/or PM2.5 are verified using a mouse model [132]. In their study, Cho et al., demonstrated that the gut microbiome contributes to sex-related differences in ozone-induced airway hyperresponsiveness, likely resulting from sex-related differences in response to short-chain fatty acids [132].

## 7. Cross-Interaction of Sex with Comorbidities or Vitamin D in Asthma Pathogenesis

Pulmonary comorbidities and middle/lower respiratory tract disorders often complicate disease management and affect patient outcomes in severe asthma [133]. Novelli et al., demonstrated that obesity is an independent factor associated with poor asthma control, while chronic rhinosinusitis with nasal polyposis is associated with airway eosinophilia [134]. Especially in the predictive factors of severe asthma in female patients, Fahem et al., indicate the existence of obesity, aspirin intolerance, gastro esophageal reflux (GERD) symptoms, and ventilator disorder at spirometry [135]. Furthermore, GERD is deeply associated with lower FEV1 and reduced FEV1/FVC in women [135]. Similarly, sex-related differences in the risk of comorbidities related to high-dose oral corticosteroid therapy in severe asthma have also been reported [136]. On the contrary, neither the comorbidities nor characteristics, including sex, influenced the treatment response to mepolizumab in patients with severe eosinophilic asthma [137]. Nevertheless, the sex-related differential pattern of comorbidity prevalence along with corticosteroid therapy has been shown to be associated with clinical costs per person [136]. Furthermore, female reproductive factors and vitamin D levels are risk factors for developing bronchiectasis [138,139]. Studies have also shown that the co-existence of bronchiectasis is associated with the clinical and radiological characteristics of patients with type 2-severe asthma [140,141].

## 8. Conclusions

Since sex affects the outcomes of medical studies, researchers should consider reporting sex differences in disease pathogenesis [142]. Recent studies have attempted to classify asthma phenotypes based on clinical, immunological, and pathological features [21]. To date, regarding the female-predominant asthma phenotype, the role of sex hormones in the alteration of immune cells, comprising an immunological cascade in the development of asthma pathogenesis, has gained interest. 

In this narrative review, we have demonstrated how sex influences asthma onset or exacerbation directly or in association with other risk factors, such as psychological stress, respiratory infection, upper airway inflammation, genetic factors, exposure to ozone or air pollution, and comorbidities (Figure 2). Especially when we think of severe pathological features in female patients with asthma, we cannot ignore the co-existence of sex-related comorbidities, including obesity, aspirin intolerance, and GERD. However, there is insufficient evidence regarding the relationship between sex and other risk factors for asthma pathogenesis, such as exercise, alcohol consumption, smoking, weather, and curative drugs. Notably, recent studies have demonstrated the relationship between sex and genetic factors in the pathogenesis of asthma. Therefore, further pathophysiological research related to SNPs in candidate genes is needed to fully investigate the “cross-talk” between sex and other factors to advance the mechanism that constitutes the female-predominant asthma phenotype, leading to the understanding of the pathophysiology of sex-related asthma phenotypes and providing improved preventive measures and treatments to female patients with asthma.

## Figures and Tables

**Figure 1 biomolecules-12-00650-f001:**
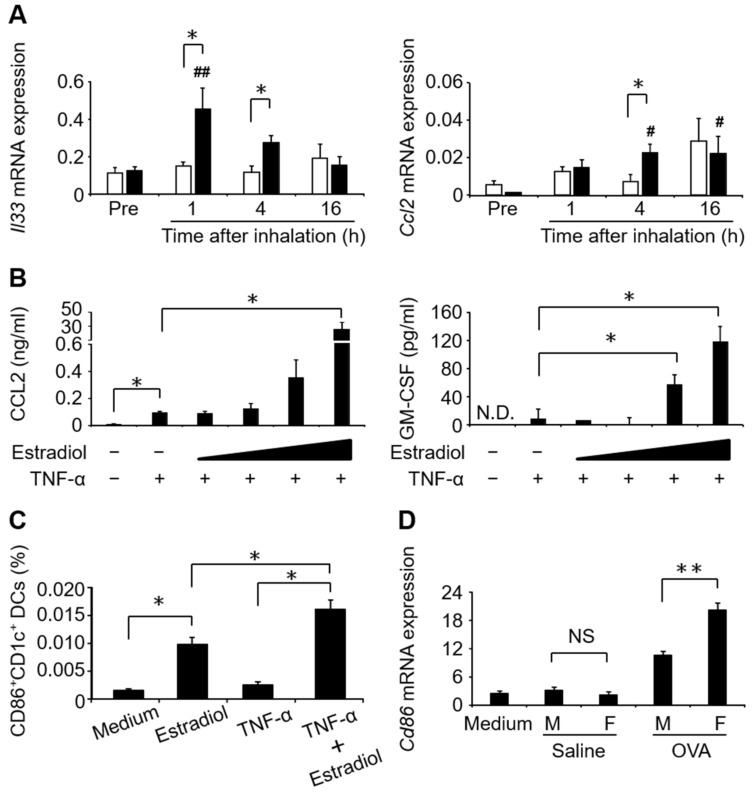
Sex-based differences in airway epithelial cell function. (**A**) Sex differences in *Il33* and *Ccl2* mRNA levels in Ep-CAM^+^ airway epithelial cells after ovalbumin (OVA) inhalation. White bars, male mice; black bars, female mice; * *p* < 0.05 vs. male mice; # *p* < 0.05 vs. before OVA inhalation (Pre); ## *p* < 0.01 vs. before OVA inhalation (Pre). Data are shown as the mean ± standard error of the mean of three mice. (**B**) GM-CSF and CCL2 production from a human epithelial cell line, BEAS-2B, after the stimulation with TNF-α in the presence or absence of 17β-estradiol for 2 d. * *p* < 0.05. Data are presented as the mean ± standard deviation (SD) of triplicate cultures. (**C**) The proportion of CD86^+^ CD1c^+^ cells among CD14^+^ cells after incubation with supernatants collected in experiment B. * *p* < 0.05. Data are presented as the mean ± SD of triplicate cultures. (**D**) The expression levels of *Cd86* mRNA in the cultured CD11c^+^ cells with BAL fluids prepared 2 d after OVA or saline inhalation. Data are presented as the mean ± SD of triplicate cultures. M, male mice; F, female mice; ** *p* < 0.01; NS, not significant. OVA, ovalbumin; *Hrpt*, hypoxanthine-guanine phosphoribosyltransferase; TNF-α, tumor necrosis factor-alpha; CCL2, chemokine (C-C motif) ligand 2; DC, dendritic cell; BAL, bronchoalveolar lavage; PBS, phosphate-buffered saline; IL33, interleukin 33; CD86, cluster of differentiation 86; RT-PCR, real-time reverse transcription-polymerase chain reaction; GM-CSF, granulocyte-macrophage colony-stimulating factor.

**Figure 2 biomolecules-12-00650-f002:**
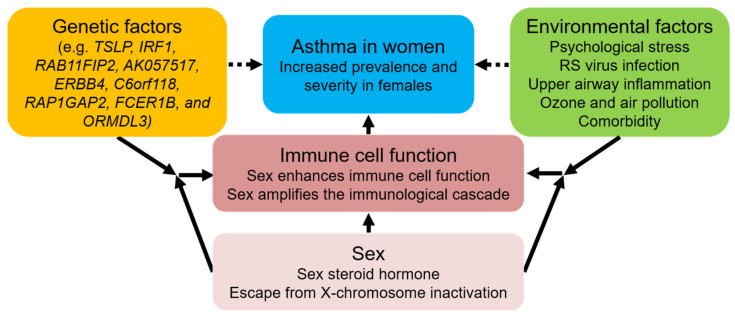
Schematic diagram illustrating the role of sex in the exacerbation of allergic asthma. Sex is associated with worse clinical outcomes of allergic asthma, including relatively higher prevalence, hospitalization rate, and relapse rate. This association is partly mediated by sex hormone-related alteration of innate and adaptive immune cells. Furthermore, in the immunological cascade from sensing allergens to airway cells, including epithelial cells and dendritic cells, to effector cell accumulation in surrounding airways, their responses are amplified based on sex. Meanwhile, the association of sex with other environmental factors such as psychological stress, respiratory infection, and upper airway inflammation induces sex-related dysmorphism of immune responses in asthma. In addition, the interplay between sex and genetic factors such as *TSLP* SNP rs2289276 or rs1837253 may also contribute to female-specific asthmatic immune responses. Thus, sex plays a multifaced role in female-dominant asthma pathogenesis. TSLP, thymic stromal lymphopoietin; SNP, single nucleotide polymorphism; RS, respiratory syncytial virus.

## Data Availability

Not applicable.

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
