# Peer review of "Sex Plays a Multifaceted Role in Asthma Pathogenesis"

_biomolecules, 2022, doi:10.3390/biom12050650_

Round 1

Reviewer 1 Report

The review presented by the authors entitled "Sex plays a multifaceted role in asthma pathogenesis" is very interesting and highlights a very current thermostatics related to sex genger in the context of asthma, and to the risks associated with the severity of the disease and factors genetic that can influence the various outcomes.
However, some important data already present in the literature are not described which deserve to be added to make the work much more attractive for readers and the scientific community.
1) The importance and role of comorbidities have not been highlighted. In particular, it would be useful to insert a paragraph relating to sex female and comorbidities in the asthmatic patient.
2) Furthermore, in recent years, much attention has been paid to the role of bronchiectasis and vitamin D, especially in the female sex. Regarding vitamin D, it is documented that a deficiency often correlates with the severity of asthma and with the outcomes of the disease.
In this regard, insert in paritcolare the following bibliographical references.
- Rogliani P, Sforza M, Calzetta L. The impact of comorbidities on severe asthma. Curr Opin Pulm Med. 2020; 26 (1): 47-55. doi: 10.1097 / MCP.000000000000064;
- Novelli F, Bacci E, Latorre M, et al. Comorbidities are associated with different features of severe asthma. Clin Mol Allergy. 2018; 16: 25. Published 2018 Dec 3. doi: 10.1186 / s12948-018-0103-x;
- Crimi C, Ferri S, Campisi R, Crimi N. The Link between Asthma and Bronchiectasis: State of the Art. Respiration. 2020; 99 (6): 463-476. doi: 10.1159 / 000507228;
- Ferri S, Crimi C, Heffler E, Campisi R, Noto A, Crimi N. Vitamin D and disease severity in bronchiectasis. Respir Med. 2019; 148: 1-5. doi: 10.1016 / j.rmed.2019.01.009;
- Crimi C, Campisi R, Nolasco S, et al. Type 2-High Severe Asthma with and without Bronchiectasis: A Prospective Observational Multicentre Study. J Asthma Allergy. 2021; 14: 1441-1452. Published 2021 Nov 30. doi: 10.2147 / JAA.S332245;
- Crimi C, Campisi R, Cacopardo G, et al. Real-life effectiveness of mepolizumab in patients with severe refractory eosinophilic asthma and multiple comorbidities. World Allergy Organ J. 2020; 13 (9): 100462. Published 2020 Sep 18. doi: 10.1016 / j.waojou.2020.100462

-Barry LE, O'Neill C, Patterson C, Sweeney J, Price D, Heaney LG. Age and Sex Associations with Systemic Corticosteroid-Induced Morbidity in Asthma. J Allergy Clin Immunol Pract. 2018;6(6):2014-2023.e2. doi:10.1016/j.jaip.2018.04.008

- Fahem N, Ben Saad A, Cheikhmhamed S, Migaou A, Joobeur S, Rouatbi N. Predictive factors of severe asthma in women. Tunis Med. 2019; 97 (8-9): 950-955.

In these works it is highlighted how asthma in real life in relation to the female sex is the object of study and is fundamental for clinical and therapeutic purposes.
Authors are advised to discuss these issues as well as inserting a paragraph in the Conclusions section.

Author Response

Reply to Reviewer #1

Reviewer 1: The review presented by the authors entitled "Sex plays a multifaceted role in asthma pathogenesis" is very interesting and highlights a very current thermostatics related to sex genger in the context of asthma, and to the risks associated with the severity of the disease and factors genetic that can influence the various outcomes. However, some important data already present in the literature are not described which deserve to be added to make the work much more attractive for readers and the scientific community.

Reply: Dear Reviewer, thank you for your time and efforts in reviewing our manuscript. We appreciate your constructive comments on our manuscript. We have carefully reviewed the comments and revised the manuscript accordingly. Below we have provided point-by-point responses to all your comments with the details of the changes made in the manuscript. The changes mentioned here are shown in a colored font in the revised manuscript.

1) The importance and role of comorbidities have not been highlighted. In particular, it would be useful to insert a paragraph relating to sex female and comorbidities in the asthmatic patient.

Reply: Thank you for your insightful suggestion. Accordingly, we have described the interaction between sex and comorbidities in a new section, “7. Cross-interaction of sex with vitamin or comorbidities in asthma pathogenesis” (pages 11–­12, lines 537–555).  

    “Pulmonary comorbidities and middle/lower respiratory tract disorders often complicate disease management and affect patient outcomes in severe asthma (Curr Opin Pulm Med. 2020 Jan;26(1):47-55.). Novelli et al. demonstrated that obesity is an independent factor associated with poor asthma control, while chronic rhinosinusitis with nasal polyposis is associated with airway eosinophilia (Clin Mol Allergy. 2018; 16: 25.). Especially in the predictive factors of severe asthma in female patients, Fahem et al. indicate the existence of obesity, aspirin intolerance, gastro esophageal reflux (GERD) symptoms, and ventilator disorder at spirometry (Tunis Med. 2019; 97 (8-9): 950-955.). Furthermore, GERD is deeply associated with lower FEV1 and reduced FEV1/FVC in women (Tunis Med. Aug-Sep 2019;97(8-9):950-955.). Similarly, sex-related differences in the risk of comorbidities related to high-dose oral corticosteroid therapy in severe asthma have also been reported (J Allergy Clin Immunol Pract. 2018;6(6):2014-2023.e2.). On the contrary, neither the comorbidities nor characteristics, including sex, influenced the treatment response to mepolizumab in patients with severe eosinophilic asthma (World Allergy Organ J. 2020; 13 (9): 100462.). Nevertheless, the sex-related differential pattern of comorbidity prevalence along with corticosteroid therapy has been shown to be associated with clinical costs per person (J Allergy Clin Immunol Pract. 2018;6(6):2014-2023.e2.).”

2) Furthermore, in recent years, much attention has been paid to the role of bronchiectasis and vitamin D, especially in the female sex. Regarding vitamin D, it is documented that a deficiency often correlates with the severity of asthma and with the outcomes of the disease. In this regard, insert in paritcolare the following bibliographical references.

- Rogliani P, Sforza M, Calzetta L. The impact of comorbidities on severe asthma. Curr Opin Pulm Med. 2020; 26 (1): 47-55. doi: 10.1097 / MCP.000000000000064;

- Novelli F, Bacci E, Latorre M, et al. Comorbidities are associated with different features of severe asthma. Clin Mol Allergy. 2018; 16: 25. Published 2018 Dec 3. doi: 10.1186 / s12948-018-0103-x;

- Crimi C, Ferri S, Campisi R, Crimi N. The Link between Asthma and Bronchiectasis: State of the Art. Respiration. 2020; 99 (6): 463-476. doi: 10.1159 / 000507228;

- Ferri S, Crimi C, Heffler E, Campisi R, Noto A, Crimi N. Vitamin D and disease severity in bronchiectasis. Respir Med. 2019; 148: 1-5. doi: 10.1016 / j.rmed.2019.01.009;

- Crimi C, Campisi R, Nolasco S, et al. Type 2-High Severe Asthma with and without Bronchiectasis: A Prospective Observational Multicentre Study. J Asthma Allergy. 2021; 14: 1441-1452. Published 2021 Nov 30. doi: 10.2147 / JAA.S332245;

- Crimi C, Campisi R, Cacopardo G, et al. Real-life effectiveness of mepolizumab in patients with severe refractory eosinophilic asthma and multiple comorbidities. World Allergy Organ J. 2020; 13 (9): 100462. Published 2020 Sep 18. doi: 10.1016 / j.waojou.2020.100462

-Barry LE, O'Neill C, Patterson C, Sweeney J, Price D, Heaney LG. Age and Sex Associations with Systemic Corticosteroid-Induced Morbidity in Asthma. J Allergy Clin Immunol Pract. 2018;6(6):2014-2023.e2. doi:10.1016/j.jaip.2018.04.008

- Fahem N, Ben Saad A, Cheikhmhamed S, Migaou A, Joobeur S, Rouatbi N. Predictive factors of severe asthma in women. Tunis Med. 2019; 97 (8-9): 950-955.

In these works it is highlighted how asthma in real life in relation to the female sex is the object of study and is fundamental for clinical and therapeutic purposes.

Authors are advised to discuss these issues as well as inserting a paragraph in the Conclusions section.

Reply: Thank you for highlighting this point and providing the references. We agree that our paper lacked a description of the relationship between sex and vitamin D/comorbidities. Following the suggested references, we have described the findings of some recent studies that provided insights into the relationship between sex and vitamin D. As mentioned in our previous comment, to incorporate these, we have added a new section (Cross-interaction of sex with vitamin or comorbidities in asthma pathogenesis) (pages 11–12, lines 537–555). In addition, we have also revised the Discussion section and Figure 2 to reflect the suggested changes.

“Furthermore, female reproductive factors and vitamin D levels are risk factors for developing bronchiectasis (Respir Med. 2019; 148: 1-5.; Biomedicines. 2022 Jan 28;10(2):303.). Studies have also shown that the co-existence of bronchiectasis is associated with the clinical and radiological characteristics of patients with type 2-severe asthma (J Asthma Allergy. 2021; 14: 1441-1452.; Respiration. 2020;99(6):463-476.).”

Reviewer 2 Report

Dear authors, thank you for your excellent and extensive review about this important topic.

Author Response

Reviewer 2: Dear authors, thank you for your excellent and extensive review about this important topic.

Reply: Thank you for your time and efforts in reviewing our manuscript. We appreciate your positive feedback on our manuscript. However, following the suggestions of the other reviewers, we have revised the manuscript. We believe that the revised manuscript has been improved significantly compared to the original one and would like to have your view on the revisions.

Reviewer 3 Report

To the Authors:

Summary

The purpose of this manuscript was to perform a systematic database search on the topic of sex and bronchial asthma. This narrative review highlighted the role of sex in relation to asthma development and progression using the latest evidence from murine and human studies. The influence of genetic and environmental factors in female-predominant asthma was also discussed. Plausible mechanisms were described in detail and in relation to asthma pathophysiology.

Comments to the authors:

In general, a good attempt was made by the authors to perform a qualitative analysis of current data on the sex-asthma link. The manuscript is well-written and the topic has been well-researched and is supported by relevant and latest data. However, there are a number of concerns that need to be addressed.  If this paper is a narrative review as opposed to a systematic review of the literature, then this should be clearly stated in the purpose of the study. In the case of the latter, in order to provide transparency and reproducibility, more details describing the screening procedure, selection of publications, and eligibility criteria are required. Furthermore, there are a few grammatical errors that require attention.

Please refer to o the comments below.

Comment 1: Literature search:

According to the Cochrane Library Handbook (Chapter 4) on conducting systematic reviews, it is recommended that at least 3 extensive database searches are undertaken including Medline, Embase, and Central databases, whereas Bramer et al, (2017) recommends at least 4 for adequate coverage of the literature including Google scholar for grey literature.

- As outlined in the methods of PRISMA guidelines, a detailed account of the literature search should be described clearly identifying inclusion and exclusion criteria.

Indicate, which studies (observational, randomized controlled trials) were used.  Or were meta-analyses and systematic reviews only considered? This is not clear in the manuscript.

In addition, the type of studies murine, humans (in this case ages) as well as languages.

The initial number of studies identified, abstracts and full-texts read, those excluded with reasons (for example not on the topic), as well as whether reference lists were checked for relevant papers, and the final number of studies included in the review. 

These steps can be illustrated using a flowchart.

Ideally, a table of included studies summarizing study characteristics should be included at the end of the manuscript.

References:

  1. Lefebvre C, et al. Chapter 4: Searching for and selecting studies. In: Higgins JPT, Thomas J, Chandler J, Cumpston M, Li T, Page MJ, Welch VA (editors). Cochrane Handbook for Systematic Reviews of Interventions version 6.0 (updated July 2019). Cochrane, 2019. Available from training.cochrane.org/handbook.

    https://training.cochrane.org/handbook/current/chapter-04

  1. Bramer et al, 2017. Optimal database combinations for literature searches in systematic reviews: a prospective exploratory study. Systematic Reviews; 6:245. DOI 10.1186/s13643-017-0644-y
  2. Moher et al, 2009. Preferred Reporting Items for Systematic Reviews and Meta-Analyses: The PRISMA Statement. Ann Intern Med. 151:264-269. https://doi.org/10.7326/0003-4819-151-4-200908180-00135

-Did you consider checking the grey literature (Google Scholar) or ongoing studies in clinical trial registries?

Comment 3:

Line 52 English grammar ‘ … of which 12 million comprises female patients with asthma (age, >18 years) and 7.3 million individuals comprise adult male patients’

Better to say ……12 million comprises female patients with asthma and 7.3 million adult males.

Mention asthma prevalence statistics in Japan as compared to the US and Europe.

Comment 4

Lines 52- 53 ‘The prevalence of asthma after puberty is higher in women than in men in various countries’

Elaborate on which countries by mentioning actual data.

Comment 5

Add references at end of statements on lines 69-72, 72-73, 73-76.

Lines 69-72 : ‘In particular, the influence of sex differences on environmental factors, such as psychological stress receptivity, immune responses against viral infection, and rhinitis prevalence, have been recently elucidated (ref).’

Lines 72- 73 Moreover, the contribution rate of the specific genetic risk factors for asthma is different between male and female patients (ref).

Lines 73-76: ‘This diverse range of asthma pathophysiology is classified as a phenotype or endotype based on clinical characteristics, variety of causation, factors leading to exacerbation, and molecular pathogenesis, including the type of inflammation (ref).

Comment 6 Line 128 English? Asthma reliever prescription fill rates?

Line 128 ‘………and physician office visits and have higher asthma reliever prescription fill rates than those of male patients’

Better to say higher rates for asthma reliever medication.

Comment 7: Line 440, English? ‘Approximately 47.9% of the Japanese population aged >12 years, except for hospitalized patients, report suffering or psychological stress.’

Line 440 You mean report suffering from psychological stress?

Comment 8

Line 468-469 ‘In asthma pathogenesis before puberty, asthma prevalence is higher in men than in women, and the role of RS virus infection in immunomodulation has garnered

attention.’

You mean asthma prevalence before puberty is higher in boys than in girls or males than females rather than men and women.

Author Response

Reply to Reviewer #3

Reviewer 3: Summary

The purpose of this manuscript was to perform a systematic database search on the topic of sex and bronchial asthma. This narrative review highlighted the role of sex in relation to asthma development and progression using the latest evidence from murine and human studies. The influence of genetic and environmental factors in female-predominant asthma was also discussed. Plausible mechanisms were described in detail and in relation to asthma pathophysiology.

Reply: Dear Reviewer, thank you for your thorough review and constructive comments. We have revised the manuscript accordingly and believe that the changes will be satisfactory. Below, we have provided point-by-point responses to all your comments with the detailed changes performed in the main manuscript. The changes mentioned here are shown in a colored font in the revised manuscript.

Comments to the authors:

In general, a good attempt was made by the authors to perform a qualitative analysis of current data on the sex-asthma link. The manuscript is well-written and the topic has been well-researched and is supported by relevant and latest data. However, there are a number of concerns that need to be addressed. If this paper is a narrative review as opposed to a systematic review of the literature, then this should be clearly stated in the purpose of the study. In the case of the latter, in order to provide transparency and reproducibility, more details describing the screening procedure, selection of publications, and eligibility criteria are required. Furthermore, there are a few grammatical errors that require attention.

Reply: Thank you for your constructive feedback. As you pointed out, this paper is a narrative review but not a systematic review. Therefore, we have clearly stated this point as a critical limitation of the study (Page 3, lines 113–115 and lines 124-132). In addition, the revised manuscript was verified by a native English speaker, and the grammatical errors were fixed throughout the manuscript.

Please refer to o the comments below.

Comment 1: Literature search:

According to the Cochrane Library Handbook (Chapter 4) on conducting systematic reviews, it is recommended that at least 3 extensive database searches are undertaken including Medline, Embase, and Central databases, whereas Bramer et al, (2017) recommends at least 4 for adequate coverage of the literature including Google scholar for grey literature.

- As outlined in the methods of PRISMA guidelines, a detailed account of the literature search should be described clearly identifying inclusion and exclusion criteria. Indicate, which studies (observational, randomized controlled trials) were used. Or were meta-analyses and systematic reviews only considered? This is not clear in the manuscript. In addition, the type of studies murine, humans (in this case ages) as well as languages.

The initial number of studies identified, abstracts and full-texts read, those excluded with reasons (for example not on the topic), as well as whether reference lists were checked for relevant papers, and the final number of studies included in the review. These steps can be illustrated using a flowchart. Ideally, a table of included studies summarizing study characteristics should be included at the end of the manuscript.

References:

Lefebvre C, et al. Chapter 4: Searching for and selecting studies. In: Higgins JPT, Thomas J, Chandler J, Cumpston M, Li T, Page MJ, Welch VA (editors). Cochrane Handbook for Systematic Reviews of Interventions version 6.0 (updated July 2019). Cochrane, 2019. Available from training.cochrane.org/handbook.

    https://training.cochrane.org/handbook/current/chapter-04

Bramer et al, 2017. Optimal database combinations for literature searches in systematic reviews: a prospective exploratory study. Systematic Reviews; 6:245. DOI 10.1186/s13643-017-0644-y

Moher et al, 2009. Preferred Reporting Items for Systematic Reviews and Meta-Analyses: The PRISMA Statement. Ann Intern Med. 151:264-269. https://doi.org/10.7326/0003-4819-151-4-200908180-00135

-Did you consider checking the grey literature (Google Scholar) or ongoing studies in clinical trial registries?

Reply: Thank you very much for your important comments. This review is a narrative review but not a systematic review and does not include the grey literature (Google Scholar) and ongoing studies in clinical trial registries. Therefore, it is difficult to describe strictly the inclusion and exclusion criteria and the statements for the number of papers identified and difficult to illustrate the flowchart and depict a table regarding included studies for the review. In this review, we have discussed the relationship between sex and genetic factors or environmental factors based on the literature written in English that could be picked up from searching results in PubMed and CDC reports by using specific keywords. Therefore, the description might be biased depending on how we selected the keywords and combined these words. While we have excluded case reports as much as possible, observational studies, randomized studies, and controlled trials were included. When the conflicting opinions have been reported in the field, we assigned priority to more recent publications, meta-analyses, and systematic reviews. Furthermore, we gave priority to the results from human studies than animal studies. In human studies, we have described the ages of participants in the text when it is needed to interpret the results. Because these points could be critical limitations in this review, we have clearly described the related description in section 2. “2. Search strategy and selection criteria” (Page 3, lines 112–132).

Comment 3:

Line 52 English grammar ‘ … of which 12 million comprises female patients with asthma (age, >18 years) and 7.3 million individuals comprise adult male patients’. Better to say ……12 million comprises female patients with asthma and 7.3 million adult males.

Reply: We apologize for the grammatical errors in the descriptions. The description has been revised as follows; “, including 12 million adult (age >18 years) females and 7.3 million males.” (page 2, lines 55–56). In addition, we have verified the revised manuscript by a native English speaker and fixed the grammatical errors throughout the manuscript.

Mention asthma prevalence statistics in Japan as compared to the US and Europe.

Reply: Thank you very much for your constructive comment. According to Japanese guidelines for adult asthma 2020 (Allergol Int. 2020 Oct;69(4):519-548), the prevalence of asthma in adults is assumed to be 10% and even higher in children, which is somewhat low compared to western developed countries (Lancet. 1997 Oct;350 Suppl 2:SII1-4.). We have added the related description in the Introduction section (page 2, lines 50–52).

Comment 4:

Lines 52- 53 ‘The prevalence of asthma after puberty is higher in women than in men in various countries’. Elaborate on which countries by mentioning actual data.

Reply: Thank you for highlighting this. “Several countries” include Japan, European countries, and the US. To aid clarity, we have revised the sentence as follows; “The prevalence of asthma after puberty is higher in females than in males in various countries, including Japan, European countries, and the United States (Allergol. Int, 2020, 69(4), 519-548.), (Most Recent National Asthma Data. Available online, CDC report), (Allergy, 2012, 67(1), 91-98.).” (page 2, line 56–58).

Comment 5:

Add references at end of statements on lines 69-72, 72-73, 73-76.

Lines 69-72 : ‘In particular, the influence of sex differences on environmental factors, such as psychological stress receptivity, immune responses against viral infection, and rhinitis prevalence, have been recently elucidated (ref).’

Reply: Thank you for highlighting this. Accordingly, we have added the references (page 2, lines 74–77).

Lines 72- 73 Moreover, the contribution rate of the specific genetic risk factors for asthma is different between male and female patients (ref).

Reply: Thank you very much for highlighting the missing references. Accordingly, we have added the references (page 2, line 78).

Lines 73-76: ‘This diverse range of asthma pathophysiology is classified as a phenotype or endotype based on clinical characteristics, variety of causation, factors leading to exacerbation, and molecular pathogenesis, including the type of inflammation (ref).

Reply: Thank you for highlighting this. Accordingly, we have added the references (page 2, lines 78–81).

Comment 6:

Line 128 English? Asthma reliever prescription fill rates?

Line 128 ‘………and physician office visits and have higher asthma reliever prescription fill rates than those of male patients’. Better to say higher rates for asthma reliever medication.

Reply: We apologize for the errors and confusions. In accordance with your comment, we have revised the description as follows; “…and asthma reliever medication than male patients.” (page 3, lines 138–142).

Comment 7:

Line 440, English? ‘Approximately 47.9% of the Japanese population aged >12 years, except for hospitalized patients, report suffering or psychological stress.’. Line 440 You mean report suffering from psychological stress?

Reply: Apologies for not being clear in describing our intent. We have revised the highlighted sentence as follows; “Approximately 47.9% of the Japanese population aged >12 years, except for hospitalized patients, suffer from psychological stress.” (page 10, lines 466–468)

Comment 8:

Line 468-469 ‘In asthma pathogenesis before puberty, asthma prevalence is higher in men than in women, and the role of RS virus infection in immunomodulation has garnered attention.’. You mean asthma prevalence before puberty is higher in boys than in girls or males than females rather than men and women.

Reply: We apologize for the confusion. To aid clarity, we have revised the description as follows; “In asthma pathogenesis before puberty, asthma prevalence is higher in boys than in girls before puberty,” (page 11, lines 496–498).

Reviewer 4 Report

This manuscript by Miyasaka et al summarizes our understanding of gender-related differences in asthma, which have been observed in both experimental and clinical settings. The following comments are meant to improve the manuscript.

Figure 1a is lacking legend. What do the black and white bars represent? Female and male mice, respectively?

Figures 1c and d show increase CD86+CD1+ DCs and CD86 mRNA but the importance of CD86 was only briefly mentioned in lines 296-297. It would help the reader if there was discussion of the role of CD86 in asthma and/or in sex-related differences in asthma. Are there other markers that are important in this context and are induced by 17-beta estradiol?

The manuscript also briefly discusses the interaction of sex with genetic makeup citing two articles. There are many other studies on this particular subject matter that the authors may consider to include in the manuscript. Some of them include, but not limited to, the following: PMID: 24824216 PMID: 25817197 PMID: 29588858

Ozone and air pollution are known asthma triggers. Many studies have shown that there are sex-related differences ozone-induced asthma. This manuscript would be improved if the role of gender in ozone and/or air pollution-induced asthma is included. (Some of the relevant manuscripts include, but not limited to, the following : PMID: 22524651 PMID: 30240285 PMID: 21811617)

Author Response

Reply to Reviewer #4

Reviewer 4: This manuscript by Miyasaka et al summarizes our understanding of gender-related differences in asthma, which have been observed in both experimental and clinical settings. The following comments are meant to improve the manuscript.

Reply: Dear Reviewer, thank you for your insightful comments. We appreciate your time and efforts in reviewing our manuscript. Below, we have provided a point-by-point response to each of your comments. The changes mentioned here are shown in a colored font in the revised manuscript.

Figure 1a is lacking legend. What do the black and white bars represent? Female and male mice, respectively?

Reply: Thank you for highlighting this. We apologize for the confusion caused by the missing information in the Figure legend. Accordingly, we have added the descriptions for the white and black bars as follows: “White bars, male mice; black bars, female mice” (page 6, lines 274–275).

Figures 1c and d show increase CD86+CD1+ DCs and CD86 mRNA but the importance of CD86 was only briefly mentioned in lines 296-297. It would help the reader if there was discussion of the role of CD86 in asthma and/or in sex-related differences in asthma. Are there other markers that are important in this context and are induced by 17-beta estradiol?

Reply: We appreciate your valuable comment. In accordance with your recommendation, we have added a more detailed explanation regarding the importance of CD86 expression on DCs in the enhanced Th2 cell differentiation in female mice in section 5.2 (“5.2 Sex-related differences in the activation of DCs (page 7, lines 310–317)).”

    “Moreover, when we blocked the interaction of CD86 on DCs using the specific antibodies in IL-5 production from CD4+ T cells in co-culture experiments, the sex-related difference in IL-5 production between male and female mice was completely abolished. These results indicated that the enhanced CD86 expression on DCs caused by 17β-estradiol is responsible for the enhanced Th2 cell activation in female mice. Interestingly, the 17β-estradiol-dependent enhancement of CD86 and major histocompatibility complex-II expression is further enhanced upon IL-33 treatment.”

The manuscript also briefly discusses the interaction of sex with genetic makeup citing two articles. There are many other studies on this particular subject matter that the authors may consider to include in the manuscript. Some of them include, but not limited to, the following: PMID: 24824216 PMID: 25817197 PMID: 29588858

Reply: We appreciate your constructive comment. Following your suggestion, we have added the description regarding the interaction between sex and genetic makeup with the suggested references in section 6.1 (6.1 Interaction of sex with genetic makeup (page 10, lines 445–453).

“Genome-wide interaction studies identified six sex and ethnicity-specific asthma risk alleles in the regions of IRF1, RAB11FIP2, AK057517, ERBB4, C6orf118, and RAP1GAP2 (Hum Mol Genet. 2014 Oct 1;23(19):5251-9.). Furthermore, another genome-wide analysis employing 411 European American asthmatic cases and 297 non-asthmatic cases suggested several sex-related candidate SNPs in gene loci including TSLP (Genomics. 2015 Jul;106(1):15-22.). In a study conducted in a Pakistani population, the TT genotype in rs2583476 SNP of FCER1B gene was more prevalent in the male patients with asthma than in healthy male controls, while the CT genotype in rs11650680 SNP of ORMDL3 gene was more prevalent in female asthma patients than in healthy female controls (Asthma Res Pract. 2018 Mar 22;4:4.).”

Ozone and air pollution are known asthma triggers. Many studies have shown that there are sex-related differences ozone-induced asthma. This manuscript would be improved if the role of gender in ozone and/or air pollution-induced asthma is included. (Some of the relevant manuscripts include, but not limited to, the following : PMID: 22524651 PMID: 30240285 PMID: 21811617)

Reply: Thank you for your insightful suggestion. We agree that sex is also associated with ozone and air pollution-induced asthma pathogenesis. However, our original manuscript lacked this insight. Following your suggestion, we have added the description regarding the interaction between sex and ozone or air pollution-induced asthma with the suggested references in a new section, “Section 6.4 Interaction of sex with ozone or air pollution in asthma” (page 11, lines 521–535).

“Ozone and air pollution are known as asthma triggers, and several studies have shown sex-related differences in ozone or air pollution-induced asthma pathogenesis. A study involving 30139 Chinese children aged 3–12 years demonstrated that the effects of air pollutants such as PM10 and SO2 on asthma were stronger in male children than those in female children without an allergic predisposition (PLoS One. 2011;6(7):e22470.). In contrast, among children with an allergic predisposition, positive associations between air pollutants such as SO2, NO2, and O3 and the prevalence of clinically diagnosed asthma were higher in female children than that in male children (PLoS One. 2011;6(7):e22470.). Similarly, Glad et al. demonstrated that the relationship between ozone/PM2.5 levels and emergency department visits for asthma are influenced by gender (Arch Environ Occup Health. 2012;67(2):103-8.). These sex-related differences in the airway response to ozone and/or PM2.5 are verified using a mouse model (Am J Respir Cell Mol Biol. 2019 Feb;60(2):198-208.). In their study, Cho et al. demonstrated that the gut microbiome contributes to sex-related differences in ozone-induced airway hyperresponsiveness, likely resulting from sex-related differences in response to short-chain fatty acids (Am J Respir Cell Mol Biol. 2019 Feb;60(2):198-208.).”